# Diagnosing and treating leprosy in a non-endemic setting in a national centre, London, United Kingdom 1995–2018

**Diana N. Lockwood**[1,2]*, **Amy McIntosh**[3], **Margaret Armstrong**[1], **Anna M. Checkley**[1],
**Stephen L. Walker**[1,2], **Angela McBride**[4]

**1** Hospital for Tropical Diseases, University College London Hospitals NHS Foundation Trust, London, United Kingdom, **2** Faculty of Infectious and Tropical Diseases, London School of Hygiene and Tropical Medicine, London, United Kingdom, **3** Liverpool Women's Hospital, Liverpool, United Kingdom, **4** Department of Global Health and Infection, Brighton and Sussex Medical School, Brighton, United Kingdom

* diana.lockwood@lshtm.ac.uk

**Data Availability Statement:** All relevant data are within the manuscript and its Supporting Information files.

## Abstract

### Background

Leprosy is rare in the United Kingdom (UK), but migration from endemic countries results in new cases being diagnosed each year. We documented the clinical presentation of leprosy in a non-endemic setting.

### Methods

Demographic and clinical data on all new cases of leprosy managed in the Leprosy Clinic at the Hospital for Tropical Diseases, London between 1995 and 2018 were analysed.

### Results

157 individuals with a median age of 34 (range 13–85) years were included. 67.5% were male. Patients came from 34 different countries and most contracted leprosy before migrating to the UK. Eighty-two (51.6%) acquired the infection in India, Sri Lanka, Bangladesh, Nepal and Pakistan. 30 patients (19.1%) acquired leprosy in Africa, including 11 from Nigeria. Seven patients were born in Europe; three acquired their leprosy infection in Africa, three in South East Asia, and one in Europe. The mean interval between arrival in the UK and symptom onset was 5.87 years (SD 10.33), the longest time to diagnosis was 20 years.

Borderline tuberculoid leprosy (n = 71, 42.0%), and lepromatous leprosy (n =, 53 33.1%) were the commonest Ridley Jopling types. Dermatologists were the specialists diagnosing leprosy most often. Individuals were treated with World Health Organization recommended drug regimens (rifampicin, dapsone and clofazimine).

### Conclusion

Leprosy is not a disease of travellers but develops after residence in an leprosy endemic area. The number of individuals from a leprosy endemic country reflect both the leprosy prevalence and the migration rates to the United Kingdom. There are challenges in

**Funding:** Payment of publication fees was supported by the Hospital for Tropical Diseases Charitable Fund. The funders had no role in study design, data collection and analysis, decision to publish, or preparation of the manuscript.

**Competing interests:** The authors have declared that no competing interests exist.

diagnosing leprosy in non-endemic areas and clinicians need to recognise the symptoms and signs of leprosy.

## Author summary

This study describes the presentation of individuals with leprosy in a non-endemic setting. They came from 34 leprosy endemic countries to the United Kingdom where they were diagnosed with leprosy. Most patients were young adults and male. The number of individuals from a leprosy endemic country reflect both the leprosy prevalence and the migration rates to the United Kingdom. The highest numbers of affected individuals in our cohort were from India, Sri Lanka, Bangladesh, Brazil, and Nigeria. The diagnosis was delayed in many patients and needed to be made by specialists. Patients were treated with World Health Organization recommended multi-drug regimens of rifampicin, dapsone and clofazimine and/or rifampicin, ofloxacin and minocycline. Clinicians in non-endemic settings need to develop and maintain skills in suspecting and diagnosing leprosy. Dedicated services are needed to provide the specialist care individuals affected by leprosy require.

## Introduction

Leprosy is a chronic granulomatous infection caused by *Mycobacterium leprae* and it is associated with stigma. Transmission is by respiratory droplet spread from untreated individuals but infectivity is low; prolonged exposure is needed for infection to take place [1]. Individuals with leprosy develop skin lesions and neurological damage. The clinical presentation is determined by the host immune response to *M. leprae* and patients can be classified by the Ridley Jopling classification which reflects the spectrum of host response [2]. Patients with high cell mediated immunity to *M. leprae* develop few lesions in skin and nerves, the tuberculoid form of leprosy, patients with borderline types (borderline tuberculoid, (BT) borderline borderline (BB) and borderline lepromatous (BL)) have some cell mediated immunity and have a variable number of skin and nerve lesions. Patients with lepromatous leprosy (LL) have no cell mediated immunity to *M. leprae* and develop widespread disease with nodules and infiltration of the skin. The incubation period for leprosy is long and variable; it is shorter for patients with tuberculoid disease (range 2–5 years), but longer for patients with lepromatous leprosy, where it may be up to 20 years [3]. 202185 new cases of leprosy were reported to the World Health Organization (WHO) in 2019; India, Brazil and Indonesia reported 80.2% of global cases [4].

Leprosy is diagnosed clinically. The three cardinal signs of leprosy are hypopigmented or red skin lesions with definite sensory loss, thickened peripheral nerves and acid fast bacilli in slit skin smears [5]. However not all patients will have a cardinal sign. Skin and nerve biopsies are important in making the diagnosis, the findings range from finding granulomas in the skin of tuberculoid patients to acid fast bacilli and diffuse histiocytic infiltrates in the skin of lepromatous patients. Recognising these patterns requires an experienced histopathologist [6,7]. Leprosy patients may present with Type 1 leprosy reactions, comprising inflammation in skin and /or nerve with nerve tenderness and loss of function [8], these may occur before presentation [9]. Erythema Nodosum Leprosum (Type 2 reaction) may complicate lepromatous leprosy, and manifests with fever, painful skin lesions and inflammation affecting bones and testes [10]. It is occasionally seen as a presentation of leprosy [11]. Clinicians are often

unfamiliar with the varied clinical presentation of leprosy. The wide range of presentations means that affected individuals are referred to different specialists including neurologists, dermatologists, rheumatologists and surgeons. The diagnosis of leprosy is often delayed in low-prevalence settings; we previously reported that it took a mean of 1.8 years for patients to be diagnosed with leprosy in the UK [12].

Treatment with WHO recommended multi-drug therapy (MDT) (rifampicin, dapsone and clofazimine) is effective with low relapse rates [13]. However, the peripheral nerve damage caused by *M. leprae* infection and leprosy reactions can lead to permanent disability. Disfigurement and disability are associated with significant stigma: both social (negative attitudes of others) and internalised stigma. This negatively impacts on quality of life [14]. Earlier recognition and treatment may prevent permanent disability and the associated stigma, therefore prompt referral to appropriate services within the UK is important to reduce the impact of the disease on patients.

Leprosy in the UK is seen in individuals who have either lived in or spent a significant period in an endemic country. Transmission in the UK has not been reported since the 1940s [15]. Leprosy is a notifiable disease in the UK and 396 new cases of leprosy were reported from 1983 to 2012 in England and Wales [16]. The UK National Health Service provides free medical care to all eligible individuals. A primary care physician will refer an individual to secondary (usually hospital-based) care for investigation, diagnosis and management of refractory or major problems.

The Hospital for Tropical Diseases (HTD), University College London Hospitals NHS Foundation Trust provides a national referral service for leprosy patients and all patients diagnosed with leprosy in England and Wales are either seen or discussed with the clinician there. Leprosy services are also provided at infectious disease centres in Liverpool and Birmingham [15,17].

While the prevalence of leprosy in the UK is low, the late diagnosis puts individuals at increased risk of life altering disability. A better understanding of the typical presentations of leprosy in the UK will help clinicians suspect and recognise leprosy to facilitate earlier diagnosis. Understanding the typical pathways to diagnosis will enable targeted educational intervention toward the secondary care specialties most frequently referred individuals with leprosy symptoms. This study reports the demographics and clinical course of a cohort of leprosy patients at a national referral centre in London, UK.

## Methods

Data were collected from the case records of patients diagnosed with leprosy at the HTD between 1st January 1995 and 13th August 2018. All were managed by the same consultant leprologist (DNL). The records included a standardised data collection form completed at diagnosis. A standardised neurological examination of nerve tenderness, motor and sensory nerve function was completed at each visit.

Data were extracted on demographics, migration and travel history, presenting symptoms, diagnostic pathway and investigations. The most likely country of leprosy acquisition was determined by examining the time spent living in leprosy endemic countries before arrival in the UK. We excluded patients who had commenced anti-microbial treatment for leprosy prior to referral to HTD.

Data were entered into an anonymised database (SPSS Version 28.0.0.0) and analysed using descriptive statistics. Ethical review was not required for retrospective analysis of anonymized data collected routinely during clinical care.

## Case definitions

The diagnosis of leprosy was made based on clinical signs and using the cardinal signs of leprosy or other clinical signs supported by histopathology.[18] The Ridley Jopling classification was used to classify the type of leprosy using the appearance of the skin lesions, the bacterial index and histopathological findings. [2] The bacteriological index (BI) documents the patient's bacterial load. Skin smears were made from dermal material obtained from small cuts into the skin in up to six sites and stained for mycobacteria with the modified Ziehl-Neelsen method. The numbers of mycobacteria per high power field were counted and expressed on a logarithmic scale of 0–6 [19,20].

WHO classifications of paucibacillary (PB) or multibacillary (MB) were assigned using the 1998 definitions [21]. Paucibacillary patients have 1–5 skin lesions and are slit skin smear negative, multibacillary patients have six or more skin lesions and/or a positive slit skin smear.

Peripheral nerve function was assessed using the tools developed in the INFIR study [22].

A patient was classified as having enlarged or tender nerves if this was present in one or more of the great auricular, radial, radial cutaneous, median, ulnar, lateral popliteal and posterior tibial nerves. Nerve tenderness was recorded as present or absent. Motor impairment was diagnosed when a patient had a 1-point change in the MRC grading scale on any of the peripheral muscles tested.

Sensation was assessed in the patients hands and feet using Semmes Weinstein monofilaments (0.05, 0.2, 4, 10 and 300 g) [23] and the worst score recorded for 3 nerves (posterior tibial, median and ulnar) on both sides. Patients had sensory impairment if the monofilament threshold was increased from the normal threshold (200 mg for the hand and 2 g for the foot) in any nerve distribution [24,25].

Leprosy Type 1 reactions were diagnosed when there was skin inflammation and/or evidence of new nerve function loss (either sensory or motor or both). Nerve tenderness could also be present [26].

Erythema nodosum leprosum reactions were diagnosed when new painful skin lesions were present. These may be accompanied by fever, malaise, bone tenderness, orchitis and iritis [22].

## Antimicrobial treatments

Individuals diagnosed with leprosy were prescribed WHO recommended MDT. Adult PB patients received a monthly dose of rifampicin 600mg and dapsone daily 100mg for 6 months. MB patients received monthly rifampicin 600mg, monthly clofazimine 300mg and dapsone 100mg and clofazimine 50 mg daily. The MDT was provided in blister packs by WHO to the pharmacy.

The duration of treatment for MB patients was 2 years or until they became smear negative. In 1998, treatment duration was changed to 12 months.

Patients who experienced adverse effects due to WHO MDT were given monthly rifampicin 600 mg, ofloxacin 400 mg and minocycline 100 mg (ROM) and from 2016 all patients were prescribed monthly ROM.

## Treatment for reactions

Patients with Type 1 reactions were treated with a 32 week of steroids starting a daily dose of 40 mg and reducing by 5mg a month. Patients with Erythema nodosum leprosum were treated with steroids initially. If they did not respond they were treated with Thalidomide 400mg nocte.

**Table 1. Sex and age at diagnosis: frequency (%) n = 157.**

| | Age (years) | | | | |
|---|---|---|---|---|---|
| | **0–18** | **19–35** | **36–65** | **Over 65** | **Total** |
| **Male** | 3 | 57 | 37 | 9 | 106 (67.5) |
| **Female** | 1 | 27 | 15 | 8 | 51 (32.5) |
| **Total** | 4 (2.5) | 84 (53.5) | 52 (33.1) | 17 (10.8) | 157 |

## Results

157 individuals were diagnosed and treated for leprosy at the HTD (155 adults and two children aged 13 and 14 years). The median age was 34 years (range 13–85 years, interquartile range 23); 10% were over 65 years. Most patients (67.5%) were male (Table 1).

### Country of presumed acquisition

Table 2 indicates the country of presumed acquisition of leprosy; 34 countries and all six WHO regions are represented in this cohort. 87 patients (55.4%) acquired their infection in the WHO South-East Asia Region. A large proportion of these patients acquired leprosy in India (n = 42), Sri Lanka (n = 20), Bangladesh (n = 12), and Nepal (n = 7). Thirty (19.1%) patients acquired leprosy in Africa, with Nigeria being the largest contributor (n = 11). Twenty patients came from the Americas with 11 from Brazil and three patients from Caribbean islands, nine patients came from the Philippines and two from China.

The majority of patients acquired leprosy in their country of birth, whether prior to arrival in the UK or on return visits. Table 3 shows the WHO region of birth of the individuals in the cohort. The child cases contracted leprosy in their home countries, Brazil 1, East Timor 1 and India (2) aged 13,14, and 18 (2). Seven patients were born in Europe; three acquired their leprosy infection in Africa, three in South-East Asia, and one in Kosovo. The latter patient lived in a community for leprosy affected individuals with their parents before moving to the UK. All these individuals lived for more than eight years in the country of acquisition prior to developing leprosy.

### Leprosy classification

**Leprosy type.** Table 4 shows the Ridley- Jopling classification for the patients. Borderline tuberculoid leprosy was the commonest type (n = 71, 42.0%), followed by lepromatous leprosy (n = 53, 33.1%), borderline lepromatous leprosy (n = 20, 12.1%) and tuberculoid leprosy (n = 12, 5.3%). 11 patients had pure neural leprosy. According to the 1998 WHO classification, 62 patients (39.5%) had PB leprosy and 95 patients (60.5%) had MB leprosy.

### Initial presentation

133 patients (84.7%) first consulted their primary care physician with symptoms, while 19 (12.1%) attended an emergency department, and two (1.4%) had abnormalities detected on health screening.

### Referral to the Leprosy Clinic at HTD

Seven patients (4.5%) were referred directly by their primary care physician. The remainder were referred by secondary care specialists including dermatologists (56%), neurologists (14%), rheumatologists (3.8%) and infectious disease physicians (2.5%). 53 patients (34.4%) consulted two or more hospital specialties before they were referred to the HTD with a diagnosis of either suspected, or histologically confirmed leprosy. (Table 5)

**Table 2. Presumed country of leprosy acquisition.**

| Country | Frequency (%) n = 157 |
|---|---|
| India | 42 (26.8) |
| Sri Lanka | 20 (12.7) |
| Bangladesh | 12 (7.6) |
| Brazil | 11 (7.0) |
| Nigeria | 11 (7.0) |
| Philippines | 9 (5.7) |
| Nepal | 7 (4.5) |
| Somalia | 4 (2.5) |
| Timor-Leste | 4 (2.5) |
| Afghanistan | 2 (1.3) |
| Angola | 2 (1.3) |
| China | 2 (1.3) |
| Ghana | 2 (1.3) |
| Guyana | 2 (1.3) |
| Jamaica | 2 (1.3) |
| Sierra Leone | 2 (1.3) |
| Bolivia | 1 (0.6) |
| Cameroon | 1 (0.6) |
| Colombia | 1 (0.6) |
| Congo | 1 (0.6) |
| Democratic Republic of the Congo | 1 (0.6) |
| Ecuador | 1 (0.6) |
| Egypt | 1 (0.6) |
| Eritrea | 1 (0.6) |
| Indonesia | 1 (0.6) |
| Kenya | 1 (0.6) |
| Kosovo | 1 (0.6) |
| Libya | 1 (0.6) |
| Mozambique | 1 (0.6) |
| Pakistan | 1 (0.6) |
| Suriname | 1 (0.6) |
| Thailand | 1 (0.6) |
| Trinidad | 1 (0.6) |
| Uganda | 1 (0.6) |
| Unknown | 5 (3.2) |

**Table 3. WHO region of birth.**

| WHO region | Frequency (%) n = 157 |
|---|---|
| South-East Asia | 84 (53.5) |
| Africa | 30 (19.1) |
| Americas | 20 (12.7) |
| Western pacific | 11 (7.0) |
| Europe | 7 (4.5) |
| Eastern Mediterranean | 5 (3.2) |
| Total | 157 |

**Table 4. Clinical features by Ridley Jopling Classification of patients at baseline. Frequency (% of column) n = 157.**

| | TT | BT | BB | BL | LL | Total |
|---|---|---|---|---|---|---|
| **Total cases** | 12 (7.64) | 71 (45.22) | 1 (0.64) | 20 (12.74) | 53 (33.76) | 157 |
| **Number of skin lesions*** | | | | | | |
| 0—Pure neural leprosy | 4 (33.3) | 5 (7.1) | 0 (0) | 1 (5.6) | 1 (2.0) | 11 (7.3) |
| 1 | 7 (58.3) | 14 (20.0) | 0 (0) | 1 (5.6) | 2 (4.1) | 24 (16.0) |
| 2–10 | 1 (8.3) | 28 (40.0) | 1 (100) | 2 (11.2) | 11 (22.4) | 43 (28.7) |
| 11–30 | 0 (0) | 17 (24.3) | 0 (0) | 7 (38.9) | 14 (28.6) | 38 (25.3) |
| 31–100 | 0 (0) | 5 (7.1) | 0 (0) | 4 (22.2) | 18 (36.7) | 27 (18.0) |
| >100 | 0 (0) | 1 (1.4) | 0 (0) | 3 (16.7) | 3 (6.1) | 7 (4.6) |
| **Any Sensory Impairment^** | 5 (41.7) | 34 (48.6) | 1 (100) | 6 (31.6) | 29 (55.8) | 75 (48.7) |
| **Number of nerves with sensory impairment^** | | | | | | |
| Mean | 0.75 | 1.3 | 4 | 0.84 | 1.77 | 1.34 |
| 1–2 | 5 (41.7) | 20 (28.6) | 0 | 5 (26.3) | 12 (40.2) | 42 (27.2) |
| 3–4 | 0 | 10 (14.3) | 1 (100) | 2 (10.5) | 11 (21.1) | 24 (15.6) |
| 5–6 | 0 | 4 (5.7) | 0 | 0 | 6 (11.5) | 10 (0.06) |
| **Any Motor Impairment**** | 4 (33.3) | 27 (38.60) | 1 (100) | 7 (35.0) | 21 (41.2) | 60 (39.2) |
| **Mean bacterial index**** | 0 | 0.45 | 0 | 3.07 | 4.10 | |
| **Enlarged Nerves§** | 6 (50.0) | 37 (52.3) | 1 (100) | 13 (68.4) | 31 (63.3) | 88 (58.3) |
| **Tender Nerves§§** | 0 | 18 (25.7) | 1 (100) | 2 (11.1) | 9 (18.0) | 32 (21.6) |
| **Type 1 reaction** | 1 (8.3) | 28 (39.4) | 1(100) | 10 (50) | 16 (30.2) | 56 (35.7) |
| **Erythema nodosum leprosum** | 0 | 0 | 1(100) | 4 (20) | 7(13.2) | 12 (7.6) |
| **Antimicrobial Treatment** | | | | | | |
| WHO Paucibacillary MDT | 9 (75) | 48 (68.6) | 0 | 0 | 0 | 57 (36.3) |
| WHO Multibacillary MDT | 1 (0.83) | 20 (28.2) | 1 (100.0) | 18 (90.0) | 44 (83.0) | 84 (53.5) |
| Monthly ROM§§§ | 1 (0.83) | 1 (0.7) | 0 | 1 (5.0) | 8 (15.1) | 11 (7.0) |
| Other | 1 (0.83) | 2 (1.4) | 0 | 1 (5.0) | 1 (0.19) | 5 (3.2) |

TT = tuberculoid leprosy, BT = borderline tuberculoid leprosy, BB = borderline borderline leprosy, BL = borderline lepromatous leprosy, LL = lepromatous leprosy

*7 patients did not have number of lesions recorded, ^complete data for 154 patients, **complete data for 153 patients, ***of 117 patients who had a slit skin smear, §complete data for 151 patients

§§complete data for 148 patients, §§§ rifampicin, ofloxacin and minocycline

## Time to diagnosis of leprosy

Fifty-one individuals (39.6%) were diagnosed with leprosy within a year of initial symptom onset, and 132 (84.0%) were diagnosed within 5 years. 20 patients (12.8%) experienced a delay between 5 and 15 years between symptom onset and diagnosis.

**Table 5. First referral specialty: frequency (%) n = 153 (data missing for 4 individuals).**

| Specialty | Number (%) |
|---|---|
| Dermatology | 88 (56.1) |
| Neurology | 22 (14.0) |
| Other | 18 (11.5) |
| Surgery | 8 (5.1) |
| Leprosy Clinic | 7 (4.5) |
| Rheumatology | 6 (3.8) |
| Infectious diseases | 4 (2.5) |

**Table 6. Time between arrival in UK and symptom onset: frequency (%) n = 146* data missing for 11 patients.**

| Time between arrival in UK and symptom onset | Frequency (%) |
|---|---|
| Before arrival | 27 (18.5) |
| Under 1 year | 24 (16.4) |
| 1–5 years | 48 (32.9) |
| > 5–10 years | 21 (14.4) |
| Over 10 years | 26 (17.8) |

## Reasons for diagnostic delay: analysis of the timeline

Table 6 illustrates the time interval between migration to the UK and onset of symptoms attributed to leprosy. Twenty-seven (18.5%) of the patients noticed symptoms prior to their arrival in the UK, some up to a decade before. Just under half of the remaining patients developed symptoms within 5 years of their arrival to the UK (n = 72, 49.3%). However, 47(32.2%) of the patients developed symptoms more than five years after their arrival in the UK. Over half of patients (n = 89, 56.7%) sought healthcare within 12 months of symptom onset, but 43 patients (27.4%) waited longer than a year after their symptoms began to consult a health professional.

Following the initial consultation in primary care, 104 (66.2%) patients were referred to a hospital specialist within three months. Twelve individuals (7.6%) were not referred to a hospital specialist until more than a year after the initial consultation.

111 patients (70.7%) were reviewed in the Leprosy Clinic within one year of first seeing a hospital specialist. A number experienced longer delays in referral to tertiary care: 11 (7%) waited 1–2 years, 16 (10.2%) waited 2–5 years and 5 (3.2%) waited 5–10 years.

## Diagnosis, disease type and treatment

The diagnosis of leprosy was either made following a skin or nerve biopsy performed by a secondary care specialist for the investigation of unexplained symptoms, or by clinical suspicion with confirmation following referral to the Leprosy Clinic at HTD. Overall, 119 patients (75.8%) patients had a biopsy performed during the diagnostic process, either before or after referral to the Leprosy Clinic. Among those who had a biopsy performed, 88 (73.9%) had histological confirmation of leprosy as a result. 117 patients (74.5%) patients had a slit skin smear performed at diagnosis. The non-diagnostic biopsies showed non-specific changes of inflammation in the skin. Of these 31 patients with no histological features in their biopsies, 19 had neg slit skin smears, 7 had positive slit skin smears (range of BI 0.2–6)

Patients with BT leprosy had the widest range in number of lesions, with 14 (20.0%) having one lesion and 6 (8.5%) having more than 31 lesions. The patients with lepromatous leprosy had multiple lesions. The mean bacterial indices were 0 for those with tuberculoid leprosy and 4.10 for the LL group. Nerve thickening was present in 88 (58.3%) of patients overall and was noted in all Ridley-Jopling types. Nerve tenderness was present most frequently in those with BT leprosy but was seen in all leprosy types, including patients with LL. Sensory nerve function impairment (NFI) was present in 38% of patients and motor NFI in 39.2% of patients. NFI was noted in all types of leprosy.

Sixty-eight (43.3%) patients presented with a leprosy reaction. Fifty-six (35.7%) had a Type 1 reaction and 12 (7.6%) had ENL.

Fifty-seven (36.3%) patients were prescribed the WHO PB regimen, these were patients with tuberculoid leprosy and BT leprosy. Patients receiving the MB regimen were those with smear positive BT leprosy (30% of the BT patients), BB, BL and LL. Patients received their MB

treatments for variable lengths of time as the WHO treatment recommendations for leprosy altered over time, initially until smear negative as per the 1982 WHO recommendation [27], then a 24 month fixed duration regimen [28] and then treated for 12 months fixed duration after 1998 [29]. 11 patients were given monthly Rifampicin, Ofloxacin and minocycline, no adverse effects of ROM were recorded.

## Discussion

This cohort of patients with leprosy shows the demographics, common presenting features and diagnostic paths for patients diagnosed at the HTD, London, UK between 1995 and 2018. Patients acquired their leprosy in 34 different countries, so patients can present with leprosy from any region endemic for the disease. The numbers of individuals diagnosed and treated at the Leprosy Clinic at HTD in London reflect the levels of leprosy transmission in the endemic country and patterns of migration to the UK. This patient cohort included highly skilled workers including health care workers and engineers.

The predominance of patients from south Asia (India, Sri Lanka, Bangladesh, Nepal) reflects the high burden of disease in these countries. India remains the highest burden country globally with 114 451 new cases in 2019 [4]. Sri Lanka continues to have a high leprosy burden with 1 658 new cases in 2019. WHO lists the 23 high leprosy burden countries where 95% of global cases occur and patients from 12 of these countries were seen among this cohort in London [4].

The numbers of leprosy patients from a country reflects migration patterns over many years prior, from endemic to reporting country. Patients in our cohort came from 14 different African countries, with a predominance from Nigeria, reflecting the large community of Nigerians living in the UK. Nigeria reported 2424 cases of leprosy in 2020. Brazil is the second highest leprosy burden country globally and 11 patients were from Brazil. There were no patients from Vietnam in our cohort, unlike Canada where 20% of the patients in one cohort came from Vietnam [30], during 1979–2002. Leon *at al* report from Atlanta, USA and found 6.7% of their patients were from Vietnam in 2002–14 [31].

We cannot be certain when or where the individuals in this cohort acquired leprosy, although the majority were born and lived in their country of presumed acquisition. Many returned for visits between their arrival in the UK and development of symptoms. Our data show that leprosy is not a disease of travellers acquired after a short exposure in an endemic country. All the patients born in non-endemic countries spent at least eight years living in leprosy endemic regions.

Leprosy has a long incubation period, and 18% of individuals had an interval of more than 10 years between leaving the endemic country and diagnosis of leprosy. This extended duration between migration and diagnosis has also been reported in the Liverpool cohort, with one patient presenting 38 years after leaving a leprosy endemic area [15]. Clinicians should be aware of this long incubation period.

Patients had all types of Ridley- Jopling classification of leprosy. Borderline tuberculoid (BT) leprosy type was present in 42% of cases. This group had the largest range of skin lesions with 8.5% having more than 30 skin lesions and 25.7% had tender nerves indicating active neural inflammation. In India BT leprosy was also the commonest Ridley-Jopling type in a cohort of MB patients [32]. This is clinically important, because BT patients can develop nerve damage rapidly and need to be warned about reactions and the development of new nerve damage. They and their families can be reassured that they are of very low infectivity. Lepromatous leprosy (LL) was present in 33.1% of our patients, with 63.3% having nerve enlargement. This is a common finding in settings where diagnosis is delayed. Patients with LL are at

a higher risk of ENL reactions and managing ENL is a major challenge [33]. 7% of patients had pure neural leprosy, leprosy without skin lesions. This is a difficult type of leprosy to diagnose because it requires a nerve biopsy to demonstrate the histological appearances consistent with *M. leprae* infection. Peripheral nerve biopsy is only available in specialist centres and has mainly been reported from India and Brazil. Our close links with The National Hospital for Neurology and Neurosurgery, Queen Square, London, facilitate diagnosis and referral of patients with suspected pure neural leprosy.

Considering nerve function Impairment 38.3% patients had sensory and 39.2% had motor impairment at diagnosis; these are high rates and may reflect the late diagnosis in some patients in our cohort. In the INFIR study in India, 21% patients had new sensory loss and 15.8% had recent motor loss [22]. The impact of lasting nerve function impairment is likely to have a disproportionate impact on leprosy patients, and many suffered unemployment and financial hardship due to their leprosy related disability.

A significant proportion (42.7%) of patients presented in reaction, 35.7% with Type 1 reaction and 7.6% with ENL. Other studies have found that over 33% of patients with borderline leprosy present with Type 1 reactions [34]. Much smaller numbers of patients present with ENL reactions [35].

Adverse effects of MDT included severe haemolysis caused by dapsone. Patients are anxious about the disclosure of their leprosy diagnosis by visible clofazimine skin pigmentation, which increases stigma.

Our analysis of the referral pathway showed that delays at two major stages were common. Firstly, delayed initial presentation to a healthcare professional for review of leprosy related symptoms, and secondly delayed recognition of leprosy by secondary care specialists. Only a few patients experienced a long delay in primary care, the majority being referred rapidly for specialist review. Patients referred to dermatology clinics had a shorter time to diagnosis than those attending other specialties. Dermatologists are both more likely to suspect leprosy and are also likely to undertake skin biopsy for a number of indications; leprosy may then be diagnosed by the histopathologist. Histopathological misdiagnoses such as cutaneous tuberculosis may also occur.

One of the shortcomings of this study was that we did not collect systematic data on eye involvement and cannot report on that aspect of the patient presentation. Eye care is provided by a specialist ophthalmologist who reviews the leprosy patients regularly.

These findings demonstrate the importance of a national specialist leprosy clinic, with a role in early diagnosis, treatment and management of disability and, critically, education of doctors from a wide range of specialities who may need to recognize a clinical case of leprosy. Our clinic comprises a dedicated physician, a team of nurses with expertise in wound care and access to physiotherapists, specialist footwear and occupational therapists. A specialist pharmacy team support the prescribing and dispensing of MDT and other medications such as thalidomide. A pregnancy prevention programme is needed to manage prescribing thalidomide safely in patients with ENL. Our multi-drug therapy is supplied by the WHO. Liaison with a neurologist is often essential, both in making a diagnosis of leprosy and also in excluding the diagnosis. The histopathologist is key; the diagnosis of leprosy is often made when a biopsy is taken [7]. The service needs supporting with a dedicated ophthalmologist and surgeon to manage the eyes and the complications individuals develop in neuropathic limbs. Patients with neuropathic pain were managed jointly with the Pain Management clinic at Chelsea and Westminster Hospital. We work closely with UK Health Security Agency (previously Public Health England) for case notification and management of contacts of notified leprosy patients. Patients need psychological support to cope with the different aspects of their diagnosis, including stigma and disability, and holistic support when managing issues such as housing,

employment and immigration status. This is particularly important in this group of patients who often experience significant stigma because of their diagnosis. The service also supports colleagues managing patients with leprosy in other parts of the country.

This retrospective cohort shows that leprosy continues to present in non-endemic settings due to the persistence of the disease in many countries.

The long incubation period means that leprosy will continue to develop in individuals at risk for many years to come. Dermatologists and neurologists need to have training and continued medical education about leprosy so that the diagnosis is considered. A referral pathway and an opportunity to discuss potential cases with experts remains a vital component of good quality care for individuals diagnosed with leprosy.

## Acknowledgments

We are grateful to the patients who attended the Leprosy Clinic over many years. It was a privilege to be part of the team looking after these patients with their disease that required looking after them medically and managing their disease and their reactions. They often needed support to manage the stigma that they experienced. Many healthcare professionals have helped and supported the Leprosy Clinic team at The Hospital for Tropical Diseases, University College London Hospitals. The nursing care was provided Miriam Henman, Anna Burnley and other members of the nursing team. The pharmacy services were provided by June Minton. Many trainee medical staff worked in the clinic. Expert Histopathology services were provided by Professor Sebastian Lucas and Dr Ula Mahadeva, St. Thomas' Hospital. Neuropathic pain services were provided by Professor Andrew Rice, Department of Pain Management, Chelsea and Westminster Hospital.

Valentina de Sario helped with extracting data from case notes and data entry.

## Author Contributions

**Conceptualization:** Diana N. Lockwood, Anna M. Checkley.

**Data curation:** Margaret Armstrong, Angela McBride.

**Formal analysis:** Amy McIntosh, Angela McBride.

**Methodology:** Diana N. Lockwood, Angela McBride.

**Project administration:** Margaret Armstrong.

**Supervision:** Diana N. Lockwood, Anna M. Checkley.

**Writing – original draft:** Diana N. Lockwood, Stephen L. Walker, Angela McBride.

**Writing – review & editing:** Diana N. Lockwood, Stephen L. Walker, Angela McBride.

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
