## [Decision Letter · Decision Letter 0]

15 Jun 2022

Dear Dr Lockwood,

Thank you very much for submitting your manuscript "Diagnosing and treating leprosy in a non-endemic setting in a national centre, London, United Kingdom 1995-2018" for consideration at PLOS Neglected Tropical Diseases. As with all papers reviewed by the journal, your manuscript was reviewed by members of the editorial board and by several independent reviewers. The reviewers appreciated the attention to an important topic. Based on the reviews, we are likely to accept this manuscript for publication, providing that you modify the manuscript according to the review recommendations. 

Sincerely,

Mauro Sanchez, ScD

Associate Editor

Elizabeth Batty

Deputy Editor

Reviewer's Responses to Questions

**Key Review Criteria Required for Acceptance?**

**Methods**

-Are the objectives of the study clearly articulated with a clear testable hypothesis stated?

-Is the study design appropriate to address the stated objectives?

-Is the population clearly described and appropriate for the hypothesis being tested?

-Is the sample size sufficient to ensure adequate power to address the hypothesis being tested?

-Were correct statistical analysis used to support conclusions?

-Are there concerns about ethical or regulatory requirements being met?

Reviewer #1: Sound methodology.

The Abstract says new cases were analyzed, but this is not actually stated in the Methods section of the full paper. It would be interesting to know how many other cases were seen over this time period (previously treated, true relapses, people attending to continue treatment started elsewhere, etc.), although the subject of the paper is clearly limited to new cases.

Reviewer #2: In this retrospective analysis, data was collected from 1995 to 2018 of patients diagnosed with leprosy at the HTD. Data on demographics, migration, travel, symptoms, diagnostics, treatment were collected and reported. The data is clearly presented in Tables (1: Sex and age at diagnosis, 2: Presumed country of acquisition, 3: WHO region of birth, 4: Clinic features of RJ Classification, 5: First referral specialty, 6: Time between arrival in UK and symptom onset). The methods are clearly defined and appropriate

**Results**

-Does the analysis presented match the analysis plan?

-Are the results clearly and completely presented?

-Are the figures (Tables, Images) of sufficient quality for clarity?

Reviewer #1: The paper is about diagnosis, which is covered well.

One area I think could be xpanded is the information about skin biopsies, which were done on 119 cases, but only 88 (74%) were diagnostic for leprosy. What were the results for the 31 cases that did not indicate leprosy? Were there other diagnoses, or were they mainly 'non-specific' changes? Another question of interest is whether the positivity rate changed over time - perhaps improving in later years, with better techniques, or perhaps declining due to loos of expertise? Were any biopsy results revised on further examination?

As the diagnosis in children is becoming a key indicator of transmission, I think it would be helpful to give more detail about the two child cases - where were they thought to contract leprosy? It would also be helpful to know the age of the persons diagnosed after living in Kosovo and Jamaica, which are probably now non-endemic - one would expect them to be quite elderly.

Reviewer #2: The results of the retrospective analysis are clearly presented in tables. The information is depicted in a clear and concise manner and each table is relevant to the study.

**Conclusions**

-Are the conclusions supported by the data presented?

-Are the limitations of analysis clearly described?

-Do the authors discuss how these data can be helpful to advance our understanding of the topic under study?

-Is public health relevance addressed?

Reviewer #1: The Conclusions are sound.

Reviewer #2: This is an excellent and important retrospective analysis of patients with leprosy that presented and were treated at HTD from 1995- 2018. The authors discuss that the majority of the patients acquired leprosy in their country of birth "whether prior to arrival in the UK or on return visits". Given the timing of acquisition of leprosy, it was felt that leprosy is not a disease of travellers. Overall, the conclusions appear to be well-supported by the data. I appreciate the data on time between arrival in the UK and symptom onset. Interestingly, 26% had been present in the UK for greater than 10 years. I wonder if the patients with delayed symptoms visited endemic countries for prolonged periods of time allowing for acquisition. I appreciate the author's discussion regarding the potentially long incubation period, which may also explain this interval.

The data on nerve function and number of patient's in reaction at time of diagnosis is valuable as well when patients present with leprosy as a diagnosis.

Overall, this is an excellent and well-written manuscript that has organised and presented a large of amount data regarding patients with leprosy who presented to HTD. It is an important disease to recognize in patients from endemic regions.

**Editorial and Data Presentation Modifications?**

Reviewer #1: (No Response)

Reviewer #2: This is an excellent manuscript with a substantial amount of data that is very relevant to the field of infectious disease among other fields. I recommend minor revision. With these revisions, I recommend accepting this manuscript for publication.

**Summary and General Comments**

Reviewer #1: The paper is well presented.

Reviewer #2: This is an excellent, well-written and eloquent retrospective analysis on a large and import color of patients see at HTD from 1995-2018. I want to commend the authors for the amount of data obtained and analyzed. This study highlights several important factors including potentially long incubation time to develop leprosy, diagnostics, treatment with need for multidisciplinary care. The stigma of leprosy is also addressed, which is important. This is an important study and will be a valuable contribution to the literature. I have a few small recommendations.

line 42: please correct comma location: 53, 33.1%

line 272: MDT side effects mentioned.Are there any side effects of ROM that were experienced by patients? 

Line 371: thalidomide is mentioned as a treatment. It would be interesting for the authors to list treatments that were used for type 1 reactions and ENL in addition to thalidomide. I do not think patient specific data is needed, just a general list of medications would be interesting.

PLOS authors have the option to publish the peer review history of their article (what does this mean?). If published, this will include your full peer review and any attached files.

Reviewer #1: Yes: Paul Saunderson

Reviewer #2: No

Figure Files:

Data Requirements:

Reproducibility:

References

---

## [Decision Letter · Decision Letter 1]

7 Sep 2022

Dear Dr Lockwood,

We are pleased to inform you that your manuscript 'Diagnosing and treating leprosy in a non-endemic setting in a national centre, London, United Kingdom 1995-2018' has been provisionally accepted for publication in PLOS Neglected Tropical Diseases.

Best regards,

Mauro Sanchez, ScD

Academic Editor

Elizabeth Batty

Section Editor

<style type="text/css">p.p1 {margin: 0.0px 0.0px 0.0px 0.0px; line-height: 16.0px; font: 14.0px Arial; color: #323333; -webkit-text-stroke: #323333}span.s1 {font-kerning: none

</style>

Reviewer's Responses to Questions

**Key Review Criteria Required for Acceptance?**

**Methods**

-Are the objectives of the study clearly articulated with a clear testable hypothesis stated?

-Is the study design appropriate to address the stated objectives?

-Is the population clearly described and appropriate for the hypothesis being tested?

-Is the sample size sufficient to ensure adequate power to address the hypothesis being tested?

-Were correct statistical analysis used to support conclusions?

-Are there concerns about ethical or regulatory requirements being met?

Reviewer #1: (No Response)

Reviewer #2: Excellent, well-described.

**Results**

-Does the analysis presented match the analysis plan?

-Are the results clearly and completely presented?

-Are the figures (Tables, Images) of sufficient quality for clarity?

Reviewer #1: (No Response)

Reviewer #2: (No Response)

**Conclusions**

-Are the conclusions supported by the data presented?

-Are the limitations of analysis clearly described?

-Do the authors discuss how these data can be helpful to advance our understanding of the topic under study?

-Is public health relevance addressed?

Reviewer #1: (No Response)

Reviewer #2: (No Response)

**Editorial and Data Presentation Modifications?**

Reviewer #1: (No Response)

Reviewer #2: (No Response)

**Summary and General Comments**

Reviewer #1: (No Response)

Reviewer #2: (No Response)

PLOS authors have the option to publish the peer review history of their article (what does this mean?). If published, this will include your full peer review and any attached files.

Reviewer #1: **Yes: **Paul Saunderson

Reviewer #2: No

---

## [Editor Report · Acceptance letter]

17 Oct 2022

Dear Prof. Lockwood,

We are delighted to inform you that your manuscript, "Diagnosing and treating leprosy in a non-endemic setting in a national centre, London, United Kingdom 1995-2018," has been formally accepted for publication in PLOS Neglected Tropical Diseases.

Best regards,

Shaden Kamhawi

co-Editor-in-Chief

Paul Brindley

co-Editor-in-Chief
